# Osteopontin Gene Polymorphisms Are Associated with Cardiovascular Risk Factors in Patients with Premature Coronary Artery Disease

**DOI:** 10.3390/biomedicines9111600

**Published:** 2021-11-02

**Authors:** Nonanzit Pérez-Hernández, Rosalinda Posadas-Sánchez, Gilberto Vargas-Alarcón, Lizet Paola Hernández-Germán, Verónica Marusa Borgonio-Cuadra, José Manuel Rodríguez-Pérez

**Affiliations:** 1Department of Molecular Biology, Instituto Nacional de Cardiología Ignacio Chávez, Mexico City 14080, Mexico; unicanona@yahoo.com.mx (N.P.-H.); gvargas63@yahoo.com (G.V.-A.); hgerlizlop@hotmail.com (L.P.H.-G.); 2Department of Endocrinology, Instituto Nacional de Cardiología Ignacio Chavez, Mexico City 14080, Mexico; rossy_posadas_s@yahoo.it; 3Department of Genomic Medicine, Instituto Nacional de Rehabilitación Luis Guillermo Ibarra, Mexico City 14389, Mexico; vborgoni@yahoo.com.mx

**Keywords:** osteopontin, coronary artery disease, polymorphic sites, biomineralization, metabolic abnormalities

## Abstract

Osteopontin (*OPN*) is considered a clinical predictor of cardiovascular disease. We aimed to evaluate the association of the *OPN* gene polymorphisms rs2728127 and rs11730582 with the development of premature coronary artery disease (pCAD), cardiovascular risk factors, and cardiometabolic parameters. We evaluated 1142 patients with pCAD and 1073 controls. Both polymorphisms were determined by Taqman assays. Similar allele and genotype frequencies were observed in both groups; additionally, an association of these polymorphisms with CAD and cardiometabolic parameters was observed in both groups. In patients with pCAD, the rs11730582 was associated with a high risk of hypoadiponectinemia (OR = 1.300, P _additive_ = 0.003), low risk of hypertension (OR = 0.709, P _codominant 1_ = 0.030), and low risk of having high non-HDL cholesterol (OR = 0.637, P _additive_ = 0.038). In the control group, the rs2728127 was associated with a low risk of fatty liver (OR = 0.766, P _additive_ = 0.038); while the rs11730582 was associated with a low risk of hypoadiponectinemia (OR = 0.728, P _dominant_ = 0.022), and risk of having elevated apolipoprotein B (OR = 1.400, P _dominant_ = 0.031). Our results suggest that in Mexican individuals, the rs11730582 and rs2728127 *OPN* gene polymorphisms are associated with some abnormal metabolic variables in patients with pCAD and controls.

## 1. Introduction

Coronary artery disease (CAD) is a complex and multifactorial pathology characterized by chronic vascular inflammation. In this disease, the accumulation of lipid material within the layers of arteries causes endothelium damage, and as a final consequence, the formation of atherosclerotic plaques [1,2,3]. During the progression of the disease, some plaques are calcified mainly by deposits of calcium phosphate [2,4]. Therefore, vascular calcification and coronary artery calcium (CAC) are of clinical relevance as risk markers for CAD [5,6,7]. Several studies have reported the importance of cardiac calcification, and now this process is considered an important marker to early detection of CAD, therefore useful in primary preventive care and treatment of CAD [8,9,10]. 

Osteopontin (*OPN*) is an extracellular matrix protein, with diverse functions, including growth factor, structural matrix protein, modulator of matrix-cell interactions, among others. Besides, due to the different post-translational modifications, *OPN* participate in several biological processes, mainly in the regulation of calcification, biomineralization and proinflammatory response [11,12,13,14]. The human *OPN* gene is located on chromosome 4 (4q21-23) and is formed by seven exons [13]. This protein has pleiotropic effects, and it is expressed in several cell types such as osteocytes, osteoblasts, pre-osteoblasts, macrophages, T-cells, endothelial cells and fibroblasts, and in different tissues including bone, cartilage, vascular tissue, brain and kidney, with essential physiological and pathophysiological functions [11,13,14,15]. *OPN* also plays a preponderant role in different pathologies such as autoimmune disorders, chronic inflammatory diseases, distinct types of cancer, diabetes, cardiovascular diseases, among others [15,16]. Under physiological conditions, low circulating *OPN* levels as well as low *OPN* expression in vasculature are associated with various pathologies, for instance, diabetes or non-small cell lung cancer [17,18]. Under injury conditions, *OPN* is over-expressed and upregulated in different vascular cell types including macrophages, vascular smooth muscle cells and endothelial cells [11]. Furthermore, various studies have detected increased circulating serum levels of *OPN* in cardiovascular diseases, associated particularly with severity of coronary atherosclerosis; high plasma levels of *OPN*, associated with increased risk for cardiac events; and a high expression of *OPN* gene in calcified carotid atheroma [10,19,20,21,22]. In the last decade, genetic studies have helped to identify predisposition markers in the development of CAD; the single nucleotide polymorphisms (SNPs) have been reported as genetic markers. Particularly, the *OPN* gene contains various polymorphic sites in the promoter region, but despite the important participation of the *OPN* molecule in the physiopathology of cardiovascular diseases, few association studies between this gene and vascular diseases have been performed. Mainly, there have been studies on coronary artery calcification or large artery atherosclerosis; nonetheless, there is only one previous study of *OPN* gene polymorphisms with CAD [23,24,25].

The human *OPN* gene is located on chromosome 4 (4q21-23) and is formed by seven exons [13]. This gene contains various polymorphic sites, and despite the important participation of the *OPN* molecule in the physiopathology of cardiovascular diseases, few association studies between this gene and vascular diseases have been performed. Therefore, more studies in different populations are needed to establish the possible role of this gene in the susceptibility to develop CAD.

Thus, the purpose of this research was to analyze the association of *OPN* gene polymorphisms (rs2728127 and rs11730582) with the risk of developing premature coronary artery disease (pCAD), cardiovascular risk factors, and cardiometabolic parameters in a representative sample of Mexican mestizo individuals. In addition, we performed bioinformatic analyses to predict the possible functional role of these polymorphisms.

## 2. Materials and Methods

### 2.1. Study Individuals 

This is a cross-sectional and case-control study of individuals included in the Genetics of Atherosclerotic Disease (GEA) Mexican study. We included 2215 unrelated participants (1142 patients with pCAD and 1073 healthy controls). All participants were consecutively recruited during 2008 to 2013. For the present research, we considered as pCAD the incidence of a clinical coronary event before 55 years of age in men and 65 years in women [26,27]. The patient group was integrated by individuals diagnosed with pCAD established through a previous clinical history of myocardial infarction, angioplasty, revascularization surgery, or through the presence of coronary stenosis >50%, within the age limits indicated above.

In the control group, we included individuals without a personal or family history of CAD. Individuals from the blood bank at the Instituto Nacional de Cardiología Ignacio Chavez (INCICh) in Mexico were invited to participate as controls. Additionally, through brochures posted in social service centers in Mexico City and the INCICh, more individuals were gathered. The exclusion criteria in this group were: thyroid disorders, renal or heart failure, oncological and liver diseases.

### 2.2. Clinical, Demographic, Biochemical Variables Assessment

Demographic, clinical and biochemical data of participants from the GEA study had been previously reported [28,29,30,31,32,33]. Briefly, body mass index (BMI) was obtained as: weight (kg)/height (m^2^), waist circumference was obtained in cm, and obesity was defined when the BMI was ≥30 kg/m^2^. Type 2 diabetes mellitus (T2DM) was defined according to the American Diabetes Association criteria, with a fasting glucose ≥126 mg/dL and was also considered when participants informed the use of hypoglycemic drug treatment. Hypertension was considered when values of diastolic and systolic blood pressures were ≥90 mmHg and ≥140 mmHg, respectively, or when individuals were using antihypertensive drugs. Glucose, apolipoprotein B (ApoB), apolipoprotein A-_I_, lipoprotein (a) and lipid profiles were quantified using conventional enzymatic colorimetric techniques in a Hitachi model 902 autoanalyzer (Hitachi LTD, Tokio, Japan). Hypoalphalipoproteinemia was considered when HDL-cholesterol <50 mg/dL in women, and <40 mg/dL in men. Smoking habits were determined when the individual self-reported a current use of cigarettes. 

The high-sensibility C-reactive protein was measured by immunonephelometry (BN ProSpec Nephelometer, Date Behring, Marburgo, Germany). Levels of adiponectin were quantified with ELISA protocols (Quantikine ELISA, R&D Systems Inc., Minneapolis, MN, USA); specifically, hypoadiponectinemia was defined when adiponectin concentration was ≤25th percentile (5.30 μg/mL in men and 8.67 μg/mL in women). High alkaline phosphatase concentrations were defined when values were ≥75th percentile (83.00 IU/L in men and 90.25 IU/L in women). Percentile values of adiponectin and alkaline phosphatase were acquired from a representative subsample of the GEA participants [34,35].

### 2.3. Tomographic Evaluation

Visceral abdominal adipose tissue (VAT), liver and spleen attenuation, as well as coronary artery calcium (CAC) were quantified using the computed axial tomography system of 64 channel multidetector (Somaton Sensations, Siemens, Malvern, PA, USA). Expert radiologists measured and interpreted the scans. Scans were read to determine CAC scores using the Agatston method [36]. The specific area of VAT was determined following the description by Kvist et al. [37]. The liver to spleen attenuation ratio was determined in accordance with the report by Longo et al. [38]. Individuals in the control group with evidence of subclinical atherosclerosis were excluded from this study (CAC with a value greater than zero).

### 2.4. Estimation of Ancestry

All GEA project participants included (cases and controls) were unrelated and were ethnically matched and of self-reported Mexican mestizo ancestry. A Mexican mestizo is defined as someone born in Mexico, who is descendant from the original autochthonous inhabitants of the region and/or from Caucasian and/or African origin, who came to America during the sixteenth century. Further, we determined the genetic background of the GEA participants on the basis of a 265-ancestry markers (AIMs) panel that distinguishes Amerindian, European, and African ancestries [39]. This was assessed with specialized software and the mean global ancestry did not show significant differences in the population analyzed (55.8% vs. 54.0% Amerindian ancestry, 34.3% vs. 35.8% Caucasian ancestry and 9.8% vs. 10.1% African ancestry for patients and controls respectively, with a *p* > 0.05). Therefore, all GEA participants had a similar genetic background and thus, the population stratification was no genetic bias in the present research [40].

### 2.5. SNPs Selection and Genetic Characterization 

For this research, we chose polymorphic sites with a minor allele frequency (MAF) of at least 5% percent reported in the International HapMap Project. Furthermore, the chosen polymorphisms have been previously associated with some cardiovascular pathologies and are located in the promotor region of the *OPN* gene.

Genomic DNA was obtained from peripheral blood leukocyte samples using the QIAamp DNA blood extraction kit (Qiagen, Germany). We analyzed two polymorphisms: rs2728127, ID: C___1840806_10, sequence AAATTTTGTTGTTTTTAGAATTTTC[A/G]GACTTCCCTCCACTAAATTGACAAC, and rs11730582, ID: C___1840808_20, sequence GAGTAGTAAAGGACAGAGGCAAGTT[T/C]TCTGAACTCCTTGCAGGCTTGAAC. Both polymorphisms were determined using Taqman genotyping probes on a thermal cycler (ABI Prism 7900-Real Time) according to the manufacture’s conditions (ThermoFisher Scientific, Foster City, CA, USA). 

### 2.6. Bioinformatic Analysis

To determine the possible functional effect of *OPN* gene polymorphisms, we used the transcription factor affinity prediction (TRAP) software, available as a web server (http://trap.molgen.mpg.de/, accessed on 23 September 2021), specialized in predicting the transcription factor binding affinities to DNA sequences according to the specific polymorphisms [41].

### 2.7. Statistical Analysis

Both studied polymorphisms were in Hardy-Weinberg equilibrium (*p* > 0.05). Data are presented as median (interquartile range), mean (standard deviation) or frequencies as required. The analysis of categorical and continuous variables in the population studied was performed using chi-square, Student t, and Mann-Whitney U tests. The genetic associations of both polymorphisms with pCAD were evaluated using logistic regression analysis using the inheritance models: additive, dominant, recessive, heterozygote, codominant 1 and codominant 2. These models were adjusted by age, sex, body mass index, systolic blood pressure, diastolic blood pressure, visceral and abdominal fat, triglycerides, apolipoprotein B, serum calcium, adiponectin, and alkaline phosphatase activity. The associations of the *OPN* gene polymorphisms with cardiovascular risk factors and metabolic abnormalities were adjusted by sex, age and body mass index. All statistical analyses were performed using the SPSS software, version 24.0. Values of *p* < 0.05 were considered significant in this study.

## 3. Results

### 3.1. Evaluation of Anthropometric, Clinical and Metabolic Parameters and Prevalence of Cardiovascular Risk Factors

Anthropometric, clinical and metabolic variables of the population studied are shown in Table 1. Compared with the control group, patients with pCAD showed higher values for: body mass index, waist circumference, systolic blood pressure, visceral adipose tissue, triglycerides and glucose. In addition, prevalence of hypoalphalipoproteinemia, hypertriglyceridemia, obesity, hypertension, visceral abdominal adipose tissue and hypoadiponectinemia were also higher in patients with pCAD when compared with the control group (Table 2).

### 3.2. Association of rs2728127 and rs11730582 Polymorphisms with pCAD

A similar distribution of the two polymorphisms was observed in patients with pCAD and controls (Table 3). The inheritance models were adjusted for age, sex, body mass index, systolic and diastolic blood pressure, visceral adipose tissue, concentrations of triglycerides, apolipoprotein B, serum calcium, adiponectin and alkaline phosphatase activity.

### 3.3. Association of rs2728127 and rs11730582 Polymorphisms with Cardiovascular Risk Factors and Metabolic Parameters

The association of *OPN* gene polymorphisms with cardiovascular risk factors and metabolic parameters was assessed separately in patients with pCAD and controls. In patients with pCAD and under different inheritance models, we observed that rs11730582 was associated with an increased risk of hypoadiponectinemia (OR = 1.300, 95% CI = 1.096–1.543, P _additive_ = 0.003; OR = 1.462, 95% CI = 1.117–1.912, P _dominant_ = 0.006; OR = 1.375, 95% CI = 1.027–1.840, P _recessive_ = 0.032; OR = 1.371, 95% CI = 1.301–1.824, P _codominant 1_ = 0.031; OR = 1.678, 95% CI = 1.191–2.365, P _codominant 2_ = 0.003), with a decreased risk of hypertension (OR = 0.728, 95% CI = 0.543–0.976, P _dominant_ = 0.034; OR = 0.709, 95% CI = 0.519–0.967, P _codominant 1_ = 0.030), and low risk of having increased levels of non- HDL cholesterol (OR = 0.803, 95% CI = 0.652–0.989, P _additive_ = 0.039; OR = 0.637, 95% CI = 0.415–0.976, P _codominant 2_ = 0.038). These outcomes are depicted in Figure 1.

In the control group, the rs2728127 polymorphism was associated with a low risk of fatty liver (OR = 0.766, 95% CI = 0.596–0.985, P additive = 0.038). The rs11730582 polymorphism was associated with a lower risk of hypoadiponectinemia (OR = 0.728, 95% CI = 0.556–9.555, P dominant = 0.022; OR = 0.728, 95% CI = 0.544–0.974, P codominant 1 = 0.032), and high risk of having increased apolipoprotein B levels (OR = 1.222, 95% CI = 1.014–1.473, P additive = 0.035; OR = 1.400, 95% CI = 1.031–1.901, P dominant = 0.031; OR = 1.487, 95% CI = 1.019–2.170, P codominant 2 = 0.039). These results are shown in Figure 2. All genetic models were adjusted for age, sex, and body mass index.

## 4. Discussion

Despite the importance of *OPN* as a clinical predictor of cardiovascular disease, to date, few genetic association studies have been performed to determine its involvement as a risk marker for coronary artery disease. This genetic approach is necessary in order to have a better understanding of this complex and multifactorial disease. Thus, in this research, we evaluated two polymorphisms (rs2728127 and rs11730582) of *OPN* gene, in 1073 healthy individuals and 1142 patients with pCAD, in order to establish their association with the presence of pCAD, cardiovascular risk factors and cardiometabolic parameters in a Mexican population. This study was nested in the GEA Project, a prospective and one of the largest cohorts in Mexican population.

Our results indicate that the studied polymorphisms were not associated with the presence of pCAD. This outcome agrees with the report of Lin et al., who evaluated the rs11730582 polymorphism in 536 patients with CAD, 86 patients with peripheral artery disease (PAD) and 617 controls and did not find a significant association between this polymorphism and the presence of CAD [24]. In the same line, Hou et al., studied the association between four *OPN* gene polymorphisms (including, rs11730582) and left ventricular hypertrophy (LVH) in 1092 patients with essential hypertension; they did not find significant differences between the groups analyzed [41]. However, Jing et al., reported that rs11730582 polymorphism was associated with an increased risk of ischemic stroke [42]. The discrepancies found between the report by Jing et al. and our study could be due to different aspects: (a) Jing et al., included only 377 patients, while we analyzed 1142 individuals with pCAD; (b) Jing et al., studied patients with ischemic stroke (IS) which is a type of vascular disease, but is not specifically CAD, as in our report; (c) the study by Jing et al., was conducted in an Asian population, while our research was performed in Mexican mestizo population [43,44,45].

The *OPN* gene polymorphisms have been associated with several pathologies including Crohn’s disease [46], urolithiasis [47,48], knee osteoarthritis [49], breast cancer [50] and diabetic nephropathy [51]. As far as we know, only the study performed by Lin et al., and ours have analyzed the association between *OPN* gene polymorphisms and CAD. Any other previous research was performed evaluating non-cardiovascular pathologies. Therefore, the association of the *OPN* genetic variants with susceptibility to cardiovascular diseases remains to a great extent unknown.

We also analyzed the association of rs2728127 and rs11730582 polymorphisms with cardiovascular risk factors and metabolic parameters in both groups. In patients with pCAD, the rs11730582 polymorphism was associated with an increased risk of hypoadiponectinemia. Adiponectin is an essential protein secreted by adipocytes; currently, this adipokine is a marker for several metabolic pathologies and atherosclerotic cardiovascular disease [52,53]. For instance, Hui et al., reported that hypoadiponectinemia is related to the progression of carotid atherosclerosis [54]; while Di Chiara et al., found that hypoadiponectinemia was associated with an increased risk of left ventricular [55].

Additionally, the rs11730582 polymorphism was associated with a lower risk of hypertension, and a decreased risk of having high non-HDL-cholesterol levels. This outcome seems contradictory; nonetheless, it is important to mention that patients with pCAD included in our study received lipid-lowering and antihypertensive therapies, which probably explains these results [56,57,58,59]. Therefore, more studies are needed in this field to elucidate the complete role of this biomolecule.

In our control group, the rs2728127 polymorphism was associated with a decreased risk of having fatty liver. Evidence suggests that *OPN* plays an essential role in liver diseases [60,61], including fatty liver, which is linked to atherosclerotic cardiovascular pathology [62], as well as subclinical atherosclerosis [63]. Recently, this molecule has also been associated with the presence of cardiac arrhythmias [64]. On the other hand, the rs11730582 polymorphism was associated with a low risk of having hypoadiponectinemia. Some studies have analyzed adiponectin plasma levels in apparently healthy adults and found that hypoadiponectinemia was associated with cardiovascular risk factors leading to a pro-atherogenic condition [65,66,67].

Also in the control group, we detected an association of rs11730582 polymorphisms with a risk of having high apolipoprotein B concentrations. ApoB has been related to dyslipidemia processes and it is considered a significant predictor of cardiovascular diseases such as myocardial infarction [68]. In this sense, our control group was not under a lipid-lowering treatment, this was probably why we found an association between the rs11730582 polymorphism and elevated ApoB levels [69].

It is important to consider that genetic variation reported in other populations may not extrapolate to the Mexican genetic background, i.e., Europeans are closely related, and their genome have had fewer recombination events than Mexicans. Various genetic studies of the Mexican population have reported the proportion of Indian and White genes is 56% and 44%, respectively, in the dihybrid model and 56%, 41% and 4% from Indian, White and Blacks in the trihybrid model [44,70,71,72]. Therefore, the analysis of *OPN* polymorphisms warrants other studies in populations with different genetic backgrounds. 

Another important point to consider in the genetic variability is the Hardy-Weinberg Equilibrium (HWE) in the population analyzed. However, different factors such as a small number of participants can alter the HWE, thus influencing the inappropriate distribution of genotypes in the population. For this reason, we have included an important number of patients that warrants the HWE and supports that the population of cases and controls was ethnically well-matched.

The in-silico analysis showed that variations in the rs2728127 polymorphism could generate binding sites for the heat shock transcription factor 1 (HSF1) and the Kappa-light-chain enhancer of activated B-cells (NFκB). However, both rs2728127 and rs11730582 polymorphisms produced binding sites for the tumor suppressor Trp53 (P53) and all of them were implicated in vascular disorders. HSF1 is a transcriptional factor that interacts with heat shock element (HSE), a specific regulatory element located in promoter regions of the heat shock protein (HSP) genes [73]. The abnormal expression of HSF1 can promote cardiac injury in vascular pathologies; however, HSF1 has an important role in cellular protection from stress conditions [74]. NFκB is a nuclear transcription factor that regulates differential gene expressions affecting a broad spectrum of biological reactions including inflammation states, migration, apoptosis and cell proliferation [75,76,77]. With regards to P53, this transcription factor has a critical regulatory role in several essential genes associated with transactivation and transrepression processes. P53 has been related to angiogenesis, a process with important effects in cardiovascular diseases such as heart failure, myocardial ischemia/reperfusion, and atherosclerosis among others [78].

Some limitations in our research should be taken into consideration. First, we did not measure circulating levels of *OPN* in the participants; therefore, we did not establish if there were different concentrations of this protein between groups. For this reason, the association between *OPN* gene polymorphisms and the concentration of *OPN* was not reported. Second, we could not verify with experimental evidence if the polymorphisms had a functional impact, as we just used an in-silico approach. Third, most of the coronary risk factors were different between patients and controls; even if they were considered to adjust the statistical analyses, the possibility remains of a lower contribution of *OPN* gene polymorphisms to pCAD than that reported in this study. Nonetheless, our report also has significant strengths: (1) we used a large cohort of Mexican individuals with and without pCAD, with demographic, clinical, tomographic and biochemical variables; (2) our control group only included individuals without subclinical atherosclerosis (CAC score = zero), so there was no bias in selecting the comparison group. As far as we know, this is the first study to report the associations *OPN* gene with other cardiovascular risk factors.

## 5. Conclusions

Our results suggest an association of *OPN* gene polymorphisms with metabolic abnormalities and cardiovascular risk factors in Mexican patients with pCAD and healthy controls. These findings support the participation of *OPN* polymorphisms as metabolic markers in our population.

## Figures and Tables

**Figure 1 biomedicines-09-01600-f001:**
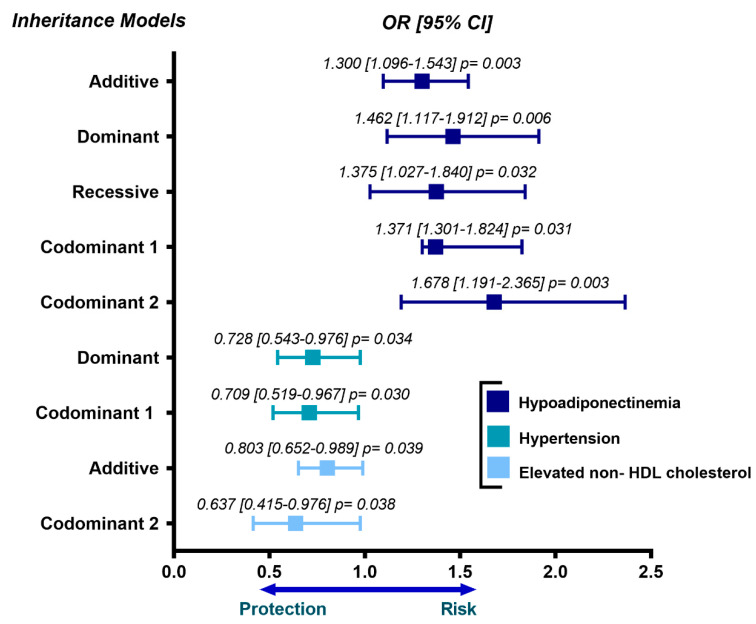
Association of rs11730582 *OPN* gene polymorphism with metabolic abnormalities in the premature coronary artery disease group. The inheritance models were adjusted for age, sex and body mass index. OR = odds ratio, CI = confidence interval.

**Figure 2 biomedicines-09-01600-f002:**
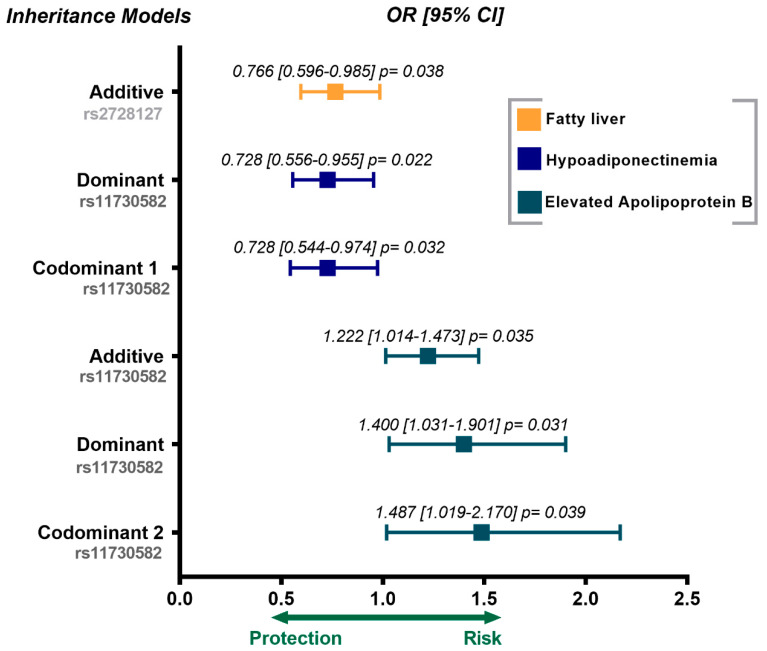
Association of *OPN* gene polymorphism with metabolic abnormalities in the control group. The inheritance models were adjusted for age, sex and body mass index. OR = odds ratio, CI = confidence interval.

**Table 1 biomedicines-09-01600-t001:** Clinical and metabolic characteristics of the individuals studied.

Characteristics	Control (*n* = 1073)	pCAD (*n* = 1142)	* *p*
Age (years)	51 ± 9	54 ± 8	<0.001
Sex (male %)	42.3	80.8	<0.001
Body mass index (kg/m2)	27.9 [25.5–30.9]	28.3 [25.9–31.1]	0.007
Waist circumference (cm)	94 ± 11	98 ± 10	<0.001
Systolic blood pressure (mmHg)	113 [104–123]	116 [106–127]	<0.001
Diastolic blood pressure (mmHg)	71 [65–77]	71 [66–78]	0.046
Visceral abdominal fat (cm2)	141 [106–181]	168 [129–218]	<0.001
High-density lipoprotein cholesterol (mg/dL)	45 [36–55]	37 [32–44]	<0.001
Low-density lipoprotein cholesterol (mg/dL)	116 [95–134]	91 [68–116]	<0.001
Triglycerides (mg/dL)	145 [108–202]	162 [119–219]	<0.001
Apolipoprotein B (mg/dL)	94 [76–113]	79 [63–102]	<0.001
Apolipoprotein A (mg/dL)	134 [115–156]	120 [101–138]	<0.001
Lipoprotein (a) (mg/dL)	5.2 [2.3–11.5]	4.8 [2.4–14.1]	0.478
Glucose (mg/dL)	90 [84–97]	95 [87–117]	<0.001
High-sensitivity C-reactive protein (mg/L)	1.5 [0.8–3.1]	1.2 [0.6–2.6]	<0.001
Adiponectin (µg/mL)	8.1 [5.0–12.8]	5.2 [3.2–8.1]	<0.001
Alkaline phosphatase (IU/L)	81 [68–96]	76 [63–95]	<0.001
Serum calcium (mg/dL)	9.7 ± 0.6	9.7 ± 0.7	0.278

Data are shown as mean ± standard deviation, median [interquartile range] or percentage. Student’s *t*-test, Mann Whitney U test or Chi-square test. pCAD = Premature Coronary Artery Disease. * *p* value.

**Table 2 biomedicines-09-01600-t002:** Prevalence of cardiovascular risk factors the individuals studied.

Characteristics	Control (*n* = 1073)	pCAD (*n* = 1142)	* *p*
LDL-cholesterol ≥ 130 mg/dL (%)	29.6	16.1	<0.001
Hypoalphalipoproteinemia (%)	52.1	66.9	<0.001
Hypertriglyceridemia (%)	47.5	56.2	<0.001
Non-HDL cholesterol > 160 mg/dL (%)	28.0	19.6	<0.001
Obesity (%)	30.5	35.0	0.024
Hypertension (%)	19.2	68.0	<0.001
High visceral abdominal adipose tissue (%)	54.8	64.5	<0.001
Current smoking (%)	22.4	11.6	<0.001
Hypoadiponectinemia (%)	42.5	57.5	<0.001
Alkaline phosphatase > p75 (%)	37.6	38.9	0.569

Data are shown as percentages. * Chi-square test. pCAD = Premature Coronary Artery Disease, LDL = low-density lipoprotein, HDL = high-density lipoprotein.

**Table 3 biomedicines-09-01600-t003:** Association of *OPN* gene polymorphisms with premature coronary artery disease.

Polymorphism	Genotype Frequency			MAF	Model	OR [95% CI]	*p*
rs2728127	*AA*	*AG*	*GG*				
					Additive	0.967 [0.804–1.164]	0.726
Control (*n =* 1073)	0.679	0.281	0.039	0.160	Dominant	1.027 [0.829–1.271]	0.807
					Recessive	0.595 [0.333–1.064]	0.080
pCAD (*n =* 1142)	0.681	0.293	0.025	0.194	Heterozygous	1.108 [0.890–1.381]	0.359
					Co-dominant 1	1.085 [0.869–1.354]	0.471
					Co-dominant 2	0.610 [0.339–1.095]	0.098
rs11730582	*CC*	*CT*	*TT*				
					Additive	0.990 [0.863–1.137]	0.891
Control (*n =* 1073)	0.301	0.469	0.247	0.464	Dominant	1.046 [0.841–1.303]	0.684
					Recessive	0.923 [0.729–1.168]	0.503
pCAD (*n =* 1142)	0.282	0.486	0.232	0.475	Heterozygous	1.100 [0.901–1.342]	0.350
					Co-dominant 1	1.159 [0.828–1.623]	0.389
					Co-dominant 2	1.288 [0.875–1.896]	0.199

The inheritance models were adjusted for age, sex, body mass index, systolic blood pressure, diastolic blood pressure, visceral abdominal fat, triglycerides, apolipoprotein B, serum calcium, adiponectin and alkaline phosphatase activity. pCAD = Premature Coronary Artery Disease, OR = Odds Ratio, CI = confidence interval, MAF= Minor Allele Frequency.

## Data Availability

The data shown in this article are available upon request from the corresponding author.

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
