# Peer review of "Osteopontin Gene Polymorphisms Are Associated with Cardiovascular Risk Factors in Patients with Premature Coronary Artery Disease"

_biomedicines, 2021, doi:10.3390/biomedicines9111600_

Round 1
Reviewer 1 Report
The manuscript “ Osteopontin (OPT) genetic variants are associated with metabolic parameters and cardiovascular factors in Mexican patients with premature coronary artery disease and healthy controls.The GEA project” is an interesting study dealing with the potential role of a novel marker of atherosclerosis in highlighting earlier the patients at risk for coronary artery disease . Although the manuscript offers many interesting elements, there are some issues that require attention and further highlighting:
- The Authors stated that they enrolled 2215 patients of which 1073 considered as healthy controls. I would like to know whether they enrolled these subjects prospectively. I suggest the authors to specify better their enrollment criteria.
- In selecting the subjects of the healthy controls group did they take into account of a risk scoring system such as Framingham-score or Euro-score. I suggest adding a brief comment.
- Looking at the Table 1, where are summarized the baseline clinical and metabolic characteristics of the population it seems that the two population have statistically different values for each parameter evaluated except for BMI, diastolic blood pressure, lipoprotein and serum calcium. Do authors have an explanation for that? Were these two groups really comparable? I suggest adding a brief comment on this.
Author Response
Reviewer 1
We thank the reviewer for the time invested in reviewing our manuscript. The observations made were very important to improve the revised version.
Comments and suggestions for authors
The manuscript “Osteopontin (OPN) genetic variants are associated with metabolic parameters and cardiovascular factors in Mexican patients with premature coronary artery disease and healthy controls. The GEA project” is an interesting study dealing with the potential role of a novel marker of atherosclerosis in highlighting earlier the patients at risk for coronary artery disease. Although the manuscript offers many interesting elements, there are some issues that require attention and further highlighting:
1) The Authors stated that they enrolled 2215 patients of which 1073 considered as healthy controls. I would like to know whether they enrolled these subjects prospectively. I suggest the authors to specify better their enrollment criteria.
Replay: Yes, the GEA project is a prospective study, with a mean follow up of 5 years. The present work had a cross-sectional design nested in the GEA project. We defined premature CAD (pCAD) as a coronary event before 55 years old for men and 65 years old for women, according to the recommendations of the ACC/AHA Guidelines on the primary Prevention of Cardiovascular Disease [26, 27]. Therefore, age for this study was limited to these age limits.
These details were emphasized in the corrected version of the manuscript as follows:
This is a cross-sectional and case-control study of individuals included in the Genetics of Atherosclerotic Disease (GEA) Mexican study. We included 2215 unrelated participants (1142 patients with pCAD and 1073 healthy controls). All participants were consecutively recruited during 2008 to 2013. For the present research, we considered as pCAD the incidence of a clinical coronary event before 55 years of age in men and 65 years in women [26, 27]. The patient group was integrated by individuals diagnosed with pCAD established through a previous clinical history of myocardial infarction, angioplasty, revascularization surgery, or through the presence of coronary stenosis >50%, within the age limits indicated above.
In the control group, we included individuals without a personal or family history of CAD. Individuals from the blood bank at the Instituto Nacional de Cardiología Ignacio Chavez (INCICh) in Mexico, were invited to participate as controls. Additionally, through brochures posted in social service centers in Mexico City and the INCICh more individuals were gathered. The exclusion criteria in this group were: thyroid disorders, renal or heart failure, oncological and liver diseases.
Additional references were added:
- Arnett, D.K.; Blumenthal, R.S.; Albert, M.A.; Buroker, A.B.; Goldberger, Z.D.; Hahn, E.J.; Himmelfarb, C.D.; Khera, A.; Lloyd-Jones, D.; Mcevoy, J.W., et al. 2019 ACC/AHA Guideline on the Primary Prevention of Cardiovascular Disease: A Report of the American College of Cardiology/American Heart Association Task Force on Clinical Practice Guidelines. Journal of the American College of Cardiology 2019, 74, e177-e232 10.1016/j.jacc.2019.03.010.
- Mahtta, D.; Ramsey, D.J.; Al Rifai, M.; Nasir, K.; Samad, Z.; Aguilar, D.; Jneid, H.; Ballantyne, C.M.; Petersen, L.A. Virani, S.S. Evaluation of Aspirin and Statin Therapy Use and Adherence in Patients With Premature Atherosclerotic Cardiovascular Disease. JAMA network open 2020, 3, e2011051 10.1001/jamanetworkopen.2020.11051.
2) In selecting the subjects of the healthy controls group did they take into account of a risk scoring system such as Framingham-score or Euro-score. I suggest adding a brief comment.
Replay: We did not consider the Framingham risk scoring system or Euro-score; we were aware to include control subjects without coronary lesions, and not to include patients with only low risk of coronary events. Therefore, the main inclusion criterion was the lack of coronary calcification documented by computed tomography (CAC score = 0).
We hope that the Reviewer agree with us that a CAC score = 0 is a better inclusion criterion for control subjects that the Framingham risk score of Euro-score.
3) Looking at the Table 1, where are summarized the baseline clinical and metabolic characteristics of the population it seems that the two population have statistically different values for each parameter evaluated except for BMI, diastolic blood pressure, lipoprotein and serum calcium. Do authors have an explanation for that? Were these two groups really comparable? I suggest adding a brief comment on this.
Replay: The Reviewer is right, most of the known coronary artery disease risk factors are out of the normal limits in patients and of course different from controls. This is an expected and unavoidable condition that was considered for the statistical analyses in order to reveal the real contribution of the OPN gene polymorphisms to pCAD. However, we recognize this issue as a weakness of this study, and following the Reviewer’s recommendation.
We included this comment in Discussion section in the corrected version of the manuscript:
Third, most of the coronary risk factors were different between patients and controls; even if they were considered to adjust the statistical analyses, it still remains the possibility of a lower contribution of OPN gene polymorphisms to pCAD than that reported in this study.

Reviewer 2 Report
Osteopontin (OPN) genetic variants are associated with metabolic parameters and cardiovascular risk factors is interesting part in for genetic researcher in CAD . It is very well analyzed and the analysis is of a high standard.
Introduction part with literature reviewing is quite interesting and well written.
This is a case-control study of individuals included in the Genetics of Atherosclerotic 73 Disease (GEA) Mexican study. I want to know “is it the first /Novel study in Mexican population or in your country”. Advise to describe more about the previous studies or related data on introduction part. According to populational data, over 50% of Mexicans can be classified as "Mestizos. " Can you please describe how to recruit the study participants section on Mexican people?
Definition on pCAD is not appropriate. Which risk calculator did you use for the assessment of pCAD in those participants? For the risk assessment of pCAD , how do you defined the Diabetes status on those population. I don’t see the HbA1c status as it is one of the strongest risk parameter for pCAD . How is their drug/medication history on assessment .
For coronary artery imaging, can you please describe more about the angiographic (invasive/noninvasive) findings with how many percent of included coronary abnormalities.
On the outcomes , data analysis was impressive and statically fantastic. The genetic variation in a population would need to be more explained, and suggested to use Hardy-Weinberg equilibrium on discussion and subgroup analysis .
Author Response
Reviewer 2
We thank the reviewer for the time invested in reviewing our manuscript. The observations made were very important to improve the revised version.
Comments and suggestions for authors
Osteopontin (OPN) genetic variants are associated with metabolic parameters and cardiovascular risk factors is interesting part in for genetic researcher in CAD. It is very well analyzed and the analysis is of a high standard.
Introduction part with literature reviewing is quite interesting and well written.
1) This is a case-control study of individuals included in the Genetics of Atherosclerotic Disease (GEA) Mexican study. I want to know “is it the first /Novel study in Mexican population or in your country”.
Replay: We appreciate the positive the Reviewer’s comments on our work. There have been of course several other attempts to study the genetics of CAD in Mexican population. However, the GEA study has the virtue to be prospective and included a large number of controls and patients. In addition, controls were selected by the absence of coronary calcification that is one of the most recognized markers of the absence of atheromas.
In order to emphasize the characteristics on the GEA project we have included the following phrases in Discussion section in the corrected version of the manuscript:
This study was nested in the GEA Project, a prospective and one of the largest cohort in Mexican population.
As far as we know, this is the first study to report the associations OPN gene with other cardiovascular risk factors.
1.1. Advise to describe more about the previous studies or related data on introduction part.
Replay: Following the Reviewer’s recommendation.
We have included additional information in the introduction section (labeled in red):
Osteopontin (OPN) is an extracellular matrix protein, with divers functions, including growth factor, structural matrix protein, modulator of matrix-cell interactions, among others. Besides, due to the different post-translational modifications OPN participate in several biological processes, mainly in the regulation of calcification, biomineralization and proinflammatory response [11-14]. The human OPN gene is located on chromosome 4 (4q21-23) and is formed by seven exons [13]. This protein has pleiotropic effects, and it is expressed in several cell types such as osteocytes, osteoblasts, pre-osteoblasts, macro-phages, T-cells, endothelial cells and fibroblasts, and in different tissues including bone, cartilage, vascular tissue, brain and kidney, with essential physiological and pathophysiological functions [15, 11, 13, 14]. OPN also plays a preponderant role in different pathologies such as autoimmune disorders, chronic inflammatory diseases, distinct types of cancer, diabetes, cardiovascular diseases, among others [16, 15]. Under physiological conditions, low circulating OPN levels as well as low OPN expression in vasculature are associated with various pathologies, for instance, diabetes or non-small cell lung cancer [17, 18]. Under injury conditions, OPN is over-expressed and upregulated in different vascular cell types including macrophages, vascular smooth muscle cells and endothelial cells [11]. Furthermore, various studies have detected increased circulating serum levels of OPN in cardiovascular diseases, associated particularly with severity of coronary atherosclerosis; high plasma levels of OPN associated with increased risk for cardiac events, and a high expression of OPN gene in calcified carotid atheroma [19-21, 10, 22]. In the last decade, genetic studies have helped to identify predisposition markers in the development of CAD; the single nucleotide polymorphisms (SNPs) have been reported as genetic makers. Particularly, the OPN gene contains various polymorphic sites in the promoter region, but despite the important participation of the OPN molecule in the physiopathology of cardiovascular diseases, few association studies between this gene and vascular diseases have been performed, mainly, has been studied in coronary artery calcification or large artery atherosclerosis, nonetheless, there is only one previous study of OPN gene polymorphisms with CAD [23-25].
Additional references were added:
- Shirakawa, K. Sano, M. Osteopontin in Cardiovascular Diseases. Biomolecules 2021, 11, 10.3390/biom11071047.
- Kahles, F.; Findeisen, H.M. Bruemmer, D. Osteopontin: A novel regulator at the cross roads of inflammation, obesity and diabetes. Molecular metabolism 2014, 3, 384-393 10.1016/j.molmet.2014.03.004.
- Eleftheriadou, I.; Tsilingiris, D.; Tentolouris, A.; Mourouzis, I.; Grigoropoulou, P.; Kapelios, C.; Pantos, C.; Makrilakis, K. Tentolouris, N. Association of Circulating Osteopontin Levels With Lower Extremity Arterial Disease in Subjects With Type 2 Diabetes Mellitus: A Cross-Sectional Observational Study. The international journal of lower extremity wounds 2020, 19, 180-189 10.1177/1534734619898097.
- Mack, P.C.; Redman, M.W.; Chansky, K.; Williamson, S.K.; Farneth, N.C.; Lara, P.N., Jr.; Franklin, W.A.; Le, Q.T.; Crowley, J.J. Gandara, D.R. Lower osteopontin plasma levels are associated with superior outcomes in advanced non-small-cell lung cancer patients receiving platinum-based chemotherapy: SWOG Study S0003. Journal of clinical oncology: official journal of the American Society of Clinical Oncology 2008, 26, 4771-4776 10.1200/jco.2008.17.0662.
- Wolak, T. Osteopontin - a multi-modal marker and mediator in atherosclerotic vascular disease. Atherosclerosis 2014, 236, 327-337 10.1016/j.atherosclerosis.2014.07.004.
- Kim, Y. Lee, C. Haplotype analysis revealed a genetic influence of osteopontin on large artery atherosclerosis. Journal of biomedical science 2008, 15, 529-533 10.1007/s11373-008-9240-4.
- Taylor, B.C.; Schreiner, P.J.; Doherty, T.M.; Fornage, M.; Carr, J.J. Sidney, S. Matrix Gla protein and osteopontin genetic associations with coronary artery calcification and bone density: the CARDIA study. Human genetics 2005, 116, 525-528 10.1007/s00439-005-1258-3.
1.3. According to populational data, over 50% of Mexicans can be classified as "Mestizos. " Can you please describe how to recruit the study participants section on Mexican people?
Replay: The Reviewer is right, most of the Mexican population is classified as “Mestizo”. This issue may influence the contribution the OPN gene polymorphisms to CAD risk factors. For this reasons, it is important to better describe the concept of “mestizo” in the manuscript as suggested by the Reviewer.
Then, we have included the following paragraph in the Materials and Methods section.
2.4. Estimation of ancestry
All GEA project participants included (cases and controls) were unrelated and were ethnically matched and of self-reported Mexican mestizo ancestry. A Mexican mestizo is defined as someone born in Mexico, who is descendant from the original autochthonous inhabitants of the region and/ or from Caucasian and/ or African origin, who came to America during the sixteenth century. Further, we determined the genetic background of the GEA participants on the basis of a 265-ancestry markers (AIMs) panel that distinguishes Amerindian, European, and African ancestries [39]. This was assessed with specialized software and the mean global ancestry did not show significant differences in the population analyzed (55.8% vs 54.0% Amerindian ancestry, 34.3% vs 35.8% Caucasian ancestry and 9.8% vs 10.1% African ancestry for patients and controls respectively, with a p > 0.05). Therefore, all GEA participants had a similar genetic background and thus, the population stratification was no genetic bias in the present research [40].
Also, these references were added in the manuscript:
- Silva-Zolezzi, I.; Hidalgo-Miranda, A.; Estrada-Gil, J.; Fernandez-Lopez, J.C.; Uribe-Figueroa, L.; Contreras, A.; Balam-Ortiz, E.; Del Bosque-Plata, L.; Velazquez-Fernandez, D.; Lara, C., et al. Analysis of genomic diversity in Mexican Mestizo populations to develop genomic medicine in Mexico. Proceedings of the National Academy of Sciences of the United States of America 2009, 106, 8611-8616 10.1073/pnas.0903045106.
- Posadas-Sánchez, R.; Pérez-Hernández, N.; Rodríguez-Pérez, J.M.; Coral-Vázquez, R.M.; Roque-Ramírez, B.; Llorente, L.; Lima, G.; Flores-Dominguez, C.; Villarreal-Molina, T.; Posadas-Romero, C., et al. Interleukin-27 polymorphisms are associated with premature coronary artery disease and metabolic parameters in the Mexican population: the genetics of atherosclerotic disease (GEA) Mexican study. Oncotarget 2017, 8, 64459-64470 10.18632/oncotarget.16223.
2) Definition on pCAD is not appropriate. Which risk calculator did you use for the assessment of pCAD in those participants?
Replay: Thanks to Reviewer for the observation, we did not really use a risk calculator, CAD patients had already suffered a coronary event when they were recruited for the study. CAD was clinically considered when patients had underwent angioplasty, revascularization surgery, or showed a coronary stenosis >50% at the moment of their inclusion in the study, as stated in Materials and Methods section.
Concerning the definition of pCAD definition of pCAD. This definition is controversial, but we followed the criteria of the ACC/AHA Guidelines on the primary Prevention of Cardiovascular Disease [26], as has been also reported by other authors [27].
We included this bibliographic support in the section of Materials and Methods as follows:
For the present research, we considered as pCAD the incidence of a clinical coronary event before 55 years of age in men and 65 years in women [26, 27]. The patient group was inte-grated by individuals diagnosed with pCAD established through a previous clinical history of myocardial infarction, angioplasty, revascularization surgery, or through the presence of coronary stenosis >50%, within the age limits indicated above.
Additional references were added:
- Arnett, D.K.; Blumenthal, R.S.; Albert, M.A.; Buroker, A.B.; Goldberger, Z.D.; Hahn, E.J.; Himmelfarb, C.D.; Khera, A.; Lloyd-Jones, D.; Mcevoy, J.W., et al. 2019 ACC/AHA Guideline on the Primary Prevention of Cardiovascular Disease: A Report of the American College of Cardiology/American Heart Association Task Force on Clinical Practice Guidelines. Journal of the American College of Cardiology 2019, 74, e177-e232 10.1016/j.jacc.2019.03.010.
- Mahtta, D.; Ramsey, D.J.; Al Rifai, M.; Nasir, K.; Samad, Z.; Aguilar, D.; Jneid, H.; Ballantyne, C.M.; Petersen, L.A. Virani, S.S. Evaluation of Aspirin and Statin Therapy Use and Adherence in Patients With Premature Atherosclerotic Cardiovascular Disease. JAMA network open 2020, 3, e2011051 10.1001/jamanetworkopen.2020.11051.
2.1 For the risk assessment of pCAD, how do you defined the Diabetes status on those population. I don’t see the HbA1c status as it is one of the strongest risk parameter for pCAD. How is their drug/medication history on assessment.
Replay: Certainly, in agreement with the Reviewer’s comment, HbA1c is one of the strongest risk parameters for pCAD in Diabetic patients. In this study, HbA1c was only quantified in individuals with T2DM for their follow-up, but it was not used for the diagnosis of diabetes. Instead, we used the ADA definition of diabetes as a fasting glucose ≥ 126 mg/dL twice or when participants informed the use of hypoglycemic drug treatment.
We have included these criteria in Materials and Methods of the corrected version of the manuscript.
Type 2 diabetes mellitus (T2DM) was defined according to the American Diabetes Associ-ation criteria, with a fasting glucose ≥ 126 mg/dL and was also considered when participants informed the use of hypoglycemic drug treatment.
Concerning hypoglycemic treatments, participants were asked for the kind of drugs and frequency they were effectively taking. However, an important proportion of diabetic patients did not remember their medicaments at the moment of this cross-sectional study. Therefore, to avoid data bias, we did not include this information in the manuscript; we hope the Reviewer agree with us in this point.
3) For coronary artery imaging, can you please describe more about the angiographic (invasive/noninvasive) findings with how many percent of included coronary abnormalities?
Replay: In the GEA study, the coronary angiography was performed only in a limited number of patients when it was required according to their medical condition in the emergency room.
However, angiographic data are not part of the parameters considered by the GEA project. Instead, we use the coronary calcification assessed by non-invasive computed tomography as the main criterion of the presence and extent of atheroma.
In this context, computed tomography of the chest was performed using a 64- channel multi-detector helical computed tomography system (Somatom Sensation, Siemens), and images were interpreted by experienced radiologists. Scans were read to determine coronary artery calcification (CAC) scores using the Agatston method.
The sentence Scans were read to determine CAC scores using the Agatston method [36]. Has been added in Material and Methods section.
This reference was added in the manuscript:
- Agatston, A.S.; Janowitz, W.R.; Hildner, F.J.; Zusmer, N.R.; Viamonte, M., Jr. Detrano, R. Quantification of coronary artery calcium using ultrafast computed tomography. Journal of the American College of Cardiology 1990, 15, 827-832 10.1016/0735-1097(90)90282-t.
4) On the outcomes, data analysis was impressive and statically fantastic. The genetic variation in a population would need to be more explained, and suggested to use Hardy-Weinberg equilibrium on discussion and subgroup analysis.
Replay: We have followed the Reviewer’s recommendation.
We have included the following paragraph in the corrected version of the manuscript:
It is important to consider that genetic variation reported in other populations, may not extrapolative to the Mexican genetic background, i.e., Europeans are closely related and their genome has had fewer recombination events than Mexicans. Various genetic studies of Mexican population have reported the proportion of Indian and White genes is 56% and 44%, respectively, in the dihybrid model and 56%, 41% and 4% from Indian, White and Blacks in the trihybrid model [70, 71, 44, 72]. Therefore, the analysis of OPN polymorphisms warrants other studies in populations with different genetic backgrounds.
Another important point to consider in the genetic variability is the Hardy-Weinberg Equilibrium (HWE) in the population analyzed. However, different factors such as a small number of participants can alter the HWE, thus influencing the inappropriate distribution of genotypes in the population. For this reason, we have included an important number of patients that warrants the HWE and supports that the population of cases and controls was ethnically well-matched.
Additional references were added:
- Lisker, R.; Ramírez, E. Babinsky, V. Genetic structure of autochthonous populations of Meso-America: Mexico. Human biology 1996, 68, 395-404
- Lisker-Yourkowitzky, R.; Ramírez-Arroyo, E.; Pérez-Rendon, G.; Díaz-Barrigo Pardo, R.; Siperstein-Blumovicz, M. Mutchinick-Baringoltz, O. Genotypes of alcohol-metabolizing enzymes in Mexicans with alcoholic liver cirrhosis. Archives of medical research 1995, 26 Spec No, S63-67
- Lisker, R.; Pérez-Briceno, R.; Granados, J. Babinsky. Gene frequencies and admixture estimates in the state of Puebla, Mexico. American journal of physical anthropology 1988, 76, 331-335 10.1002/ajpa.1330760307.
- Rosenberg, N.A.; Huang, L.; Jewett, E.M.; Szpiech, Z.A.; Jankovic, I. Boehnke, M. Genome-wide association studies in diverse populations. Nature reviews. Genetics 2010, 11, 356-366 10.1038/nrg2760.

Round 2
Reviewer 2 Report
This revised manuscript provides more information on suggested scientific facts by reviewers.
The article title is suggested to be revised as it is too long in sentence style . The title should not be in complete sentence form .
The other writing part on revision sounds good and well described .
It is impressive in that the authors describe this is the first study to report the associations OPN gene with other cardiovascular risk factors.
Author Response
Reviewer 2
We appreciate the reviewer's kind and professional support.
Comments and suggestions for authors:
This revised manuscript provides more information on suggested scientific facts by reviewers.
The article title is suggested to be revised as it is too long in sentence style. The title should not be in complete sentence form.
The other writing part on revision sounds good and well described.
It is impressive in that the authors describe this is the first study to report the associations OPN gene with other cardiovascular risk factors.
Replay: We appreciate the positive the Reviewer’s comments on our work. The Reviewer is right, the article title is too long in sentence style.
Following the recommendation of the Reviewer, we modified the title article and it was added to the final version of the manuscript.
Osteopontin gene polymorphisms are associated with cardiovascular risk factors in patients with premature coronary artery disease.
